# Periodic squeezing in a polariton Josephson junction

Albert F. Adiyatullin[1], Mitchell D. Anderson [1], Hugo Flayac [1], Marcia T. Portella-Oberli[1], Fauzia Jabeen[1], Claudéric Ouellet-Plamondon[1,3], Gregory C. Sallen[1] & Benoit Deveaud [1,2]

The use of a Kerr nonlinearity to generate squeezed light is a well-known way to surpass the quantum noise limit along a given field quadrature. Nevertheless, in the most common regime of weak nonlinearity, a single Kerr resonator is unable to provide the proper interrelation between the field amplitude and squeezing required to induce a sizable deviation from Poissonian statistics. We demonstrate experimentally that weakly coupled bosonic modes allow exploration of the interplay between squeezing and displacement, which can give rise to strong deviations from the Poissonian statistics. In particular, we report on the periodic bunching in a Josephson junction formed by two coupled exciton-polariton modes. Quantum modeling traces the bunching back to the presence of quadrature squeezing. Our results, linking the light statistics to squeezing, are a precursor to the study of nonclassical features in semiconductor microcavities and other weakly nonlinear bosonic systems.

[1] Institute of Physics, École Polytechnique Fédérale de Lausanne, 1015 Lausanne, Switzerland. [2] École Polytechnique, 91128 Palaiseau, France. [3] Present address: Niels Bohr Institute, University of Copenhagen, Blegdamsvej 17, 2100 Copenhagen, Denmark. Correspondence and requests for materials should be addressed to A.F.A. (email: albert.adiyatullin@epfl.ch) or to M.T.P-O. (email: marcia.portellaoberli@epfl.ch)

A paradigmatic manifestation of Josephson physics is the alternating particle exchange between two macroscopic quantum states subject to a potential difference, which can be observed for both fermions[1] and bosons[2, 3]. In the latter case, the Josephson dynamics can be drastically enriched by the presence of a Kerr nonlinearity[2, 4], which can give rise to such effects as self-trapping[5, 6], the unconventional photon blockade[7, 8, 9], or (spin) squeezing[10, 11]. Bosonic Josephson junctions (JJ) have been realized in various systems, including superfluid helium[12, 13], atomic condensates[5, 14], microwave photons[15], and exciton-polaritons[6, 16]. The latter quasi-particles, emerging from the strong coupling between excitons and photons in a semiconductor microcavity[17, 18], are particularly suitable for studies on the impact of the nonlinearity. It stems from the hybrid light-matter nature of the polaritons which gives rise to an effective Kerr interaction through the excitonic component, while the photonic component allows to study their emission using conventional optical means[19]. Recent progress in growth and etching techniques opens the possibility to sculpt confinement potentials seen by the polaritons at will, down to zero dimension within mesas[20] or micropillars[21]. These advances open the way to single mode Bose-Hubbard physics in a solid state system.

Interestingly, the Kerr-type nonlinearity caused by the exciton–exciton interaction is typically orders of magnitude larger than what is measured in standard nonlinear optical media. However, at the quantum level, the single particle nonlinearity $U$ remains much smaller than the mode linewidths $\kappa$ even for strong confinement. Consequently, semiconductor microcavities embody a weakly nonlinear quantum system where quantum interferences[7, 8] and quadrature squeezing[11, 22] can nevertheless be achieved towards nontrivial quantum states. While noise squeezing has been observed for exciton-polaritons[23, 24], its influence on the emission statistics has never been explored. The picosecond timescales involved in the polariton dynamics and emission events were out of the reach for the best available avalanche photodiodes which has prevented an accurate measurement of the second-order correlations. This limitation can be overcome by using a streak-camera[25–27], which acts as an ultrafast photon detector suitable for dynamical correlation measurements.

Here we demonstrate the dynamical squeezing of two populations of exciton-polaritons undergoing Josephson oscillations revealed by performing ultrafast time-resolved second-order correlation measurements. These results benefit from the nature of Josephson oscillations, which allows us to span the squeezing parameters dynamically and over a wide range. Following recent predictions[28], we show that this peculiar phenomenon is the result of the interactions between two coupled coherent states characterized by a weak nonlinearity. Capitalizing on hybrid light-matter properties of polaritons, our results demonstrate the potential to generate nonclassical light in solid state systems possessing a single particle nonlinearity like on-chip-silicon resonators[29] or microwave Josephson junctions[15].

## Results

**Polariton Josephson junction.** The JJ consists of two spatially separated polariton modes in their ground state, trapped in two artificially created circular mesas of the same size (Fig. 1a). A tunnel coupling of $J = 0.4$ meV between the two mesas results in a splitting of their ground state energies into bonding and antibonding normal modes (Fig. 1b). These states are resonantly excited with short laser pulses at an energy of $E = 1.462$ eV. The laser is focused into a 3 μm spot mostly onto one mesa, which allows us to obtain high-contrast Josephson oscillations. The light emitted by the mesas is collected in the transmission geometry

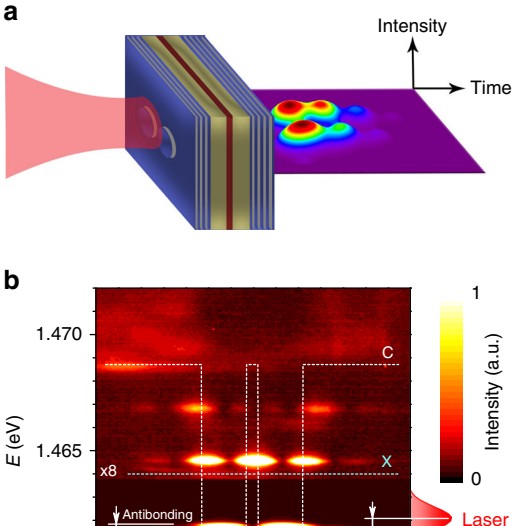

**Fig. 1** Polariton Josephson junction. **a** Schematic of the microcavity with two coupled mesas, one of which is predominantly illuminated, and the resulting Josephson oscillations of exciton-polaritons. **b** Spectrum of polariton emission from two coupled mesas under nonresonant continuous-wave excitation. Dashed curves represent the energies of the exciton (X) and cavity (C) modes. Coupling of mesas with $J = 0.4$ meV leads to formation of bonding and antibonding states. During the experiments, only these states are resonantly excited with pulsed laser, schematically shown on the right. $\Delta = \omega_c - \omega_{laser}$ is the laser detuning

with a microscope objective and sent to a beamsplitter, realizing a Hanbury Brown and Twiss (HBT) setup, with a streak-camera that acts as a single photon detector for both arms[30, 31]. The images acquired by the streak-camera are processed using a photon counting procedure. For the data presented in this paper, the statistics are accumulated for over 3.9 million laser pulses (Supplementary Note 1). By summing the photon counts arriving from all the pulses we access the dynamics of the emission intensities from the left and right mesas $I_L(t)$ and $I_R(t)$, shown in Fig. 2a. The corresponding population imbalance $z(t) = (I_L - I_R)/(I_L + I_R)$ is presented in Fig. 2b and clearly confirms the presence of Josephson oscillations of polaritons between the two mesas.

**Ultrafast time-resolved $g^2(0)$ measurements.** The second-order time correlation function is defined in the standard way

$$g^{(2)}(t_1, t_2) = \frac{\langle \hat{a}^\dagger(t_1)\hat{a}^\dagger(t_2)\hat{a}(t_2)\hat{a}(t_1)\rangle}{\langle \hat{a}^\dagger(t_1)\hat{a}(t_1)\rangle\langle \hat{a}^\dagger(t_2)\hat{a}(t_2)\rangle} \quad (1)$$

where $\hat{a}^\dagger(t)$ and $\hat{a}(t)$ are polariton creation and annihilation operators respectively. The time-dependent zero-delay second-order correlation function is defined as $g^{(2)}(0)(t) = g^{(2)}(t, t)$. Generally, $g^{(2)}(0)$ characterizes the statistics of light: it is Poissonian when $g^{(2)}(0) = 1$, super-Poissonian (bunched) when $g^{(2)}(0) > 1$, and quantum (antibunched) when $g^{(2)}(0) < 1$.

As the statistics of the polaritonic system are inherited by the photons emitted from the microcavity, we can access the second-order coherence of polaritons by counting the photon coincidences. The measured time-dependent second-order correlation function $g^{(2)}(0)(t)$ is shown in Fig. 2c, where shaded areas represent the experimental error. While the polaritonic

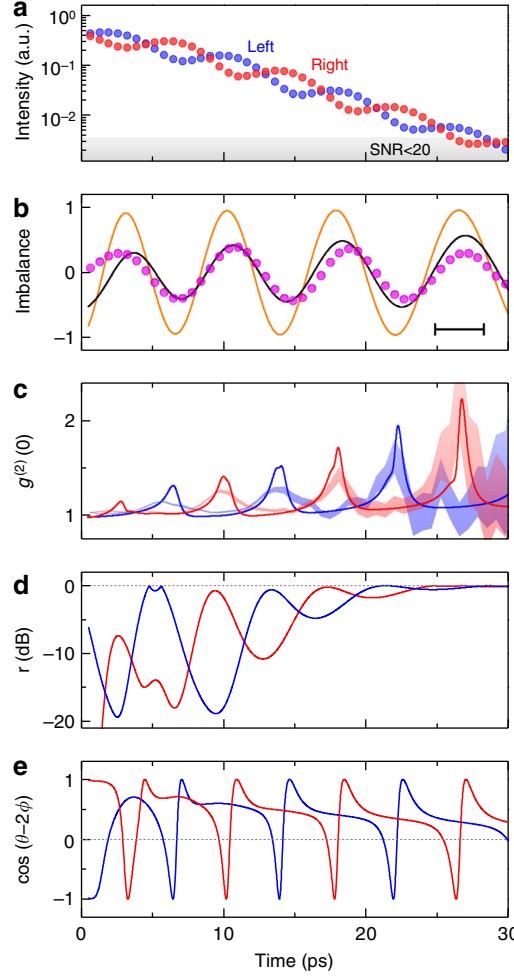

**Fig. 2** Dynamical photon bunching. **a** Measured intensity of the light emission from the left (blue) and right (red) mesa. The gray area indicates the region where signal-to-noise ratio (SNR) is insufficient for confident correlation measurements. **b** Measured population imbalance between two mesas (magenta points) clearly reveals the Josephson oscillations. Results of simulations (orange) match well the measured imbalance when convoluted with the Gaussian corresponding to the time resolution of the streak-camera being 3.4 ps (black). Time resolution is given in the plot. **c** Second-order correlation function $g^{(2)}(0)$ of the light emission from the left (blue shaded) and right (red shaded) mesa shows that the light statistics changes from Poissonian ($g^{(2)}(0) = 1$) to bunched ($g^{(2)}(0) > 1$) in phase with Josephson oscillations. Shaded areas represent the error bars calculated as the standard errors of the mean values. The corresponding results of the theoretical simulations are shown with blue and red lines. **d**, **e** Simulated evolution of **d** the absolute value of the squeezing magnitude, and **e** the cosine term from Eq. (3) for the left (blue) and right (red) mesas

populations in both mesas occupy their ground states and keep their coherence, the emission statistics clearly does not remain coherent. Indeed, $g^{(2)}(0)$ indicates that the light from each mesa changes its nature from Poissonian to bunched in phase with the Josephson oscillations. Moreover, the oscillations of $g^{(2)}(0)$ appear to be in counterphase between the two mesas, i.e., when the emission of the left mesa is bunched, the right mesa emits Poissonian light, and vice versa. Finally, the magnitude of the bunching gets higher as the polariton population decreases.

**Simulations.** We model the behavior of our system as two coupled nonlinear polariton modes with equal resonance frequency

$\omega_c$. The system Hamiltonian reads

$$\hat{\mathcal{H}} = \sum_{k=\text{L,R}} \Big[ \hbar\omega_c \hat{a}_k^\dagger \hat{a}_k + U \hat{a}_k^\dagger \hat{a}_k^\dagger \hat{a}_k \hat{a}_k + P_k(t)\hat{a}_k^\dagger + P_k^*(t)\hat{a}_k \Big] - J\Big(\hat{a}_\text{L}^\dagger \hat{a}_\text{R} + \hat{a}_\text{R}^\dagger \hat{a}_\text{L}\Big) \tag{2}$$

where $J$ is the tunneling amplitude between the two modes, and $\hat{a}_k$ are bosonic polariton operators for the fundamental trapped modes. This simplification is allowed by the resonant excitation scheme we consider, where $P_k(t)$ are the driving laser pulses specifically targeting the lowest energy normal modes. To allow for the large populations involved in our experiment, we expand the polariton operators as $\hat{a}_k = \alpha_k + \delta\hat{a}_k$, where $\alpha_k = \langle \hat{a}_k \rangle$ is the coherent mean field component and $\delta\hat{a}_k$ are the quantum fluctuation (noise) operators[22] fulfilling $\langle \delta\hat{a}_k \rangle \approx 0$. The mean-field dynamics is governed by c-number equations, whereas the fluctuation fields follow a quantum master equation accounting for interaction with the environment (see Methods). The full numerical solutions of the mean field plus fluctuation treatment are superimposed on the experimental data in Fig. 2c and show a remarkable agreement. At the same time, the measured population imbalance is well represented by the simulated one convolved with a Gaussian of FWHM = 3.4 ps, representing the experimental temporal resolution (black line in Fig. 2b).

**Two-dimensional correlation function.** More subtle features of the oscillating light statistics can be resolved when calculating the correlations between the photons arriving at different moments of time, $g^{(2)}(t_1, t_2)$ (Fig. 3a). The most salient feature of the plot are the local maxima of $g^{(2)}(t_1, t_2)$ correlation function that are arranged on a rectangular lattice. We compare the $g^{(2)}(t_1, t_2)$ plot with the numerical simulations shown in Fig. 3b and observe that the latter perfectly mimics the arrangement of the maxima of $g^{(2)}(t_1, t_2)$ in a rectangular lattice, as well as the amplitude of these maxima that increases with time. The difference in the amplitude and sharpness of these peaks results from the temporal resolution of our experiment.

## Discussion

The observed features of the light statistics arise from quadrature squeezing in a system with two coupled nonlinear states, which is sufficient to induce wide deviations to the statistics of a coherent state $|a\rangle$. Indeed, a squeezed coherent state $|\xi, \alpha\rangle = \hat{S}|\alpha\rangle$, where $\hat{S} = \exp\big[\xi^*\hat{a}^2 - \xi\hat{a}^{\dagger 2}\big]$ is the squeezing operator of complex parameter $\xi$, can demonstrate both bunching or antibunching depending on the interrelation between the amplitudes and phases of $\alpha = \bar{\alpha}e^{i\varphi}$ and $\xi = re^{i\theta}$. The second-order correlation function of such state is given by[11]

$$g^{(2)}(0) = 1 + \frac{2\bar{\alpha}^2[p - s\cos(\theta - 2\varphi)] + p^2 + s^2}{(\bar{\alpha}^2 + p)^2}, \tag{3}$$

where $p = \sinh^2(r)$ and $s = \cosh(r)\sinh(r)$, and for any value of $\alpha$ one can optimize $\xi$ to obtain a sub-Poissonian statistics. However, a sizeable non-classical statistics is restricted to small field amplitude $\bar{\alpha} \lesssim 1$. In the limit of large field, whatever the squeezing magnitude, the second-order correlation will be restricted to $1 \lesssim g^{(2)}(0) \leq 3$, which was observed, e.g., in ref. [32]. For this reason, it is far easier to reveal the squeezing for $\bar{\alpha} \gg 1$, when it manifests itself in increase of the $g^{(2)}(0)$ value (Supplementary Note 2). This is the regime we explore in our experiment carried out for large mean occupancy.

As one can see from Eq. (3), the degree of bunching depends on the relationship between the phases of the coherent state and the squeezing. This is evident from the fact that $g^{(2)}(0)$ can

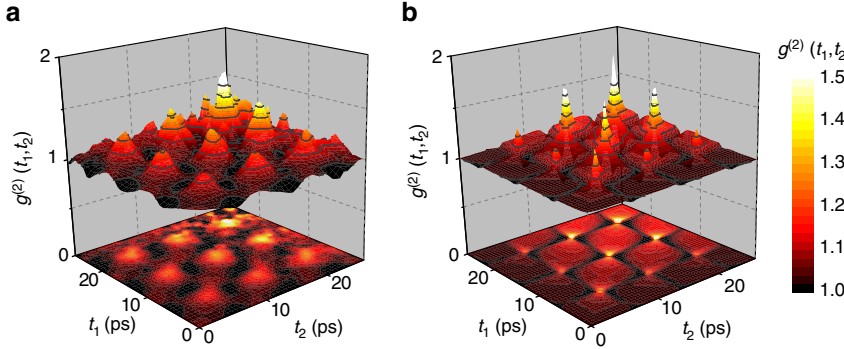

**Fig. 3** Two-dimensional correlation function. **a** Measured $g^{(2)}(t_1, t_2)$ for the emission from the left mesa. **b** Simulated $g^{(2)}(t_1, t_2)$. The arrangement of the regions where $g^{(2)}(t_1, t_2) > 1$ in a rectangular grid is well reproduced by the simulations

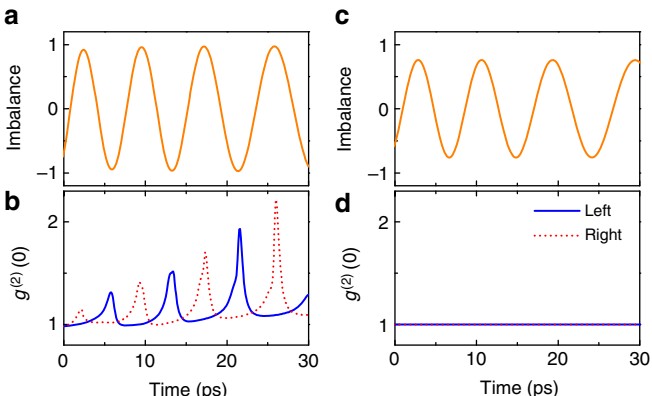

**Fig. 4** Role of nonlinearity. **a** Population imbalance and **b** second-order correlation function for interparticle interactions $U = 1.4 \, \mu eV$. **c**, **d** Same, but for $U = 0$. Other simulation parameters are given in Methods

acquire different values for $I_L = I_R$ (Fig. 2c). The calculated values of the absolute squeezing magnitude $r = |\xi|$ and relation between the phases of the displacement $\varphi$ and squeezing $\theta$ are presented in Fig. 2d, e, respectively. An interesting observation is that, in the regime of large field, the highest magnitude of squeezing does not cause the strongest bunching (Fig. 2d). The super-Poissonian light rather appears when $\cos(\theta - 2\phi)$ is negative, as it is clearly seen from Fig. 2e.

To clarify the origin of the squeezing, it is instructive to look at a linearized picture of our model. It can be obtained by omitting the higher-order terms of the Hamiltonian for the fluctuation field (7) given in the Methods section. In this framework, the terms $\alpha_k^{2*} \hat{a}_k^2 + \alpha_k^2 \hat{a}_k^{\dagger 2}$ allow us to transform the evolution equations for the fluctuation fields to those of degenerate parametric oscillators. The parametric interaction amplitude seen by the left mesa amounts to

$$\lambda_L^{\text{eff}} = U \left[ \alpha_L^2 - \frac{J^2}{U^2 \overline{\alpha}_R^4 - |\Delta_R - i\kappa/2|^2} \alpha_R^2 \right] \quad (4)$$

that we can link to a generic squeezing parameter $\xi_L$ as $\tanh(2r_L) = 2|\lambda_L^{\text{eff}}|/\kappa$ and $\theta_L = \arg(\lambda_L^{\text{eff}})$. As one can see from Eq. (4), for $J = 0$ the squeezing parameter is irrevocably bound to the displacement, and cannot be changed on demand. This clearly shows that the coupling $J$ between two modes is determinant to allow for an arbitrary control of the squeezing parameter.

The second crucial prerequisite for the manifestation of the squeezing is presence of a finite nonlinearity[28], which is also evident from the Eq. (4). To underline this, we perform simulations with $U$ set explicitly to zero (Fig. 4c, d). Even though the mean-field dynamics of Josephson oscillations can still be well described in this case, the light statistics show absolutely no deviation from Poissonian with $g^{(2)}(0) = 1$ all along the system evolution.

In summary, we have demonstrated oscillating dynamics in the statistics of light emitted from an exciton-polariton Josephson junction. We show that this behavior represents an evolution of the squeezing parameters and is a manifestation of Gaussian squeezing in coupled resonators containing weak Kerr nonlinearities. All the characteristic features of the dynamically evolving light statistics can be perfectly described within this corresponding framework. In fact, the very mechanism of Gaussian squeezing also lies at the basis of antibunching in coupled nonlinear cavities and unconventional photon blockade[7, 8] that remains elusive so far. Our results open the way towards study of this effect in similar systems[29, 33], as well as other nonclassical phenomena in the strongly correlated photonics systems[34–36].

## Methods

**Sample**. The planar microresonator consists of $\lambda$-cavity made of GaAs with a single 10 nm $In_{0.06}Ga_{0.94}As$ quantum well at an antinode of the field and sandwiched between GaAs/AlAs Bragg mirrors containing 24 and 20 pairs, respectively. It features a Rabi splitting of 3.3 meV, exciton–photon detuning of −3 meV, and a polariton lifetime of $\tau = 5.2$ ps.

For the fabrication of the coupled mesas, first, a planar half-cavity with a bottom Bragg mirror, quantum well, spacer and an etchstop was grown. Next, the mesas were fabricated by wet etching of the etchstop on a depth of 6 nm. Finally, a top Bragg mirror was grown on the top of the processed structure. Due to spatial confinement, a single mesa features a set of discrete energy levels for polaritons. For this study we used two coupled mesas with diameter of 2 μm and centre-to-centre separation of 2.5 μm leading to a coupling constant of $J = 0.4$ meV.

**Excitation scheme**. The sample is excited with the circularly polarized laser pulses generated by a Ti:Sapphire mode-locked laser in resonance with the bonding and antibonding states of the coupled mesas. To avoid excitation of the higher energy states, a pulse shaper is used to reduce the spectral width of the laser to 0.7 meV. The pulses have energy of 5 pJ. The laser emission is focused with a ×50 microscope objective into a 3 μm spot. During the experiment, the sample is held in a liquid He flow cryostat at a temperature of 5.1 K and is actively stabilized such that the excitation spot does not shift more that ≈500 nm over the course of the 34 h experiment.

**Detection scheme**. The sample emission is collected in transmission geometry using a ×50 0.42 NA microscope objective. For measuring the second-order correlation function, the sample emission is sent to the beamsplitter representing the HBT setup. Next, light from two outputs of the beamsplitter is focused on the slit of the streak-camera in synchroscan mode acting as a single photon detector. This

allows us first, to observe the photon correlations with a temporal resolution of 3.4 ps, and second, to get a real-space image of the emission. In order to isolate photons coming from a single sample excitation event, a pulse picker and an acousto-optic modulator were used to let only one laser pulse excite the sample during the streak-camera acquisition frame.

**Theoretical model.** The mean fields obey the c-number equations:

$$i\hbar\dot{\alpha}_L = \left[\Delta_L - i\kappa/2 + U|\alpha_L|^2\right]\alpha_L - J\alpha_R + P_L(t)$$
$$i\hbar\dot{\alpha}_R = \left[\Delta_R - i\kappa/2 + U|\alpha_R|^2\right]\alpha_R - J\alpha_L + P_R(t) \qquad (5)$$

where we work in the frame rotating with the laser frequency $\omega_{laser}$ and $\Delta_{L,R} = \omega_c - \omega_{laser}$ is the detuning. The modes are driven by Gaussian pulses defined as $P_{L,R}(t) = p_{L,R} \exp\left[-(t-t_0)^2/\sigma_t^2\right]$ and the relative values between $p_L$ and $p_R$ allows to mimic the position of the laser over the mesas. The fluctuation fields are governed by the master equation

$$i\hbar\frac{\partial\hat{\rho}_f}{\partial t} = \left[\hat{\mathcal{H}}_f, \hat{\rho}_f\right] - i\frac{\kappa}{2}\sum_{k=L,R}\hat{\mathcal{D}}[\delta\hat{a}_k]\hat{\rho}_f \qquad (6)$$

where $\hat{\mathcal{D}}[\hat{o}]\hat{\rho} = \{\hat{o}^\dagger\hat{o}, \hat{\rho}\} - 2\hat{o}\hat{\rho}\hat{o}^\dagger$ are standard Lindblad dissipators accounting for losses to the environment where $\kappa = \hbar/\tau$. The corresponding Hamiltonian reads

$$\begin{aligned}\hat{\mathcal{H}}_f = &\sum_{k=L,R}\left[\Delta_k\hat{a}_k^\dagger\hat{a}_k + U\left(\alpha_k^{2*}\hat{a}_k^2 + \alpha_k^2\hat{a}_k^{\dagger 2}\right)\right] \\ &+ \sum_{k=L,R}U\left[\hat{a}_k^\dagger\hat{a}_k^\dagger\hat{a}_k\hat{a}_k + 2\alpha_k^*\hat{a}_k^\dagger\hat{a}_k\hat{a}_k + 2\alpha_k\hat{a}_k^\dagger\hat{a}_k^\dagger\hat{a}_k\right] \\ &- J\left(\hat{a}_L^\dagger\hat{a}_R + \hat{a}_R^\dagger\hat{a}_L\right)\end{aligned} \qquad (7)$$

where we have omitted the $\delta$ notation for compactness. We kept here the nonlinear terms of all orders, which provides an exact quantum description. Note that the linearized picture would disregard the second line. Equations (5) and (6) are solved numerically in a Hilbert space truncated to a sufficient number of quanta to properly describe the weak fluctuation field. The expectation values are computed as $\langle\delta\hat{o} + \langle\hat{o}\rangle\hat{\mathbb{1}}\rangle = Tr\left[\left(\delta\hat{o} + \langle\hat{o}\rangle\hat{\mathbb{1}}\right)\hat{\rho}_f\right]$. The squeezing parameters $\xi_k = r_k\exp(i\theta_k)$ are extracted from

$$r_k(t) = \left[|\langle\Delta\hat{a}_k\rangle| + |\langle\hat{a}_k\rangle|^2 - \langle\hat{a}_k^\dagger\hat{a}_k\rangle\right]/2$$
$$\theta_k(t) = \arg\langle\Delta\hat{a}_k\rangle, \qquad (8)$$

where $\Delta\hat{a}_k = \langle\hat{a}_k^2\rangle - \langle\hat{a}_k\rangle^2$. The arguments of the coherent states are $\phi_k = \arg\langle\hat{a}_k\rangle$.

The simulated values of the two-time second-order correlations are obtained by summing all possible second-order truncations of the fourth-order correlations. While the third- and fourth-order correlations contribution could be added, it shows a sufficient accuracy for the large occupations we consider here. In that framework we obtain

$$g_k^{(2)}(t_1, t_2) = \frac{G_k^{(2)}(t_1, t_2)}{[N(t_1) + n_k(t_1)][N(t_2) + n_k(t_2)]} \qquad (9)$$

$$\begin{aligned}G_k^{(2)}(t_1, t_2) &= \left\langle\hat{a}_k^\dagger(t_1)\hat{a}_k^\dagger(t_2)\hat{a}_k(t_2)\hat{a}_k(t_1)\right\rangle \\ &\simeq 2\,\mathrm{Re}\left[\alpha_k(t_2)\alpha_k(t_1)\left\langle\hat{a}_k^\dagger(t_1)\hat{a}_k^\dagger(t_2)\right\rangle\right] \\ &+ 2\mathrm{Re}\left[\alpha_k^*(t_2)\alpha_k(t_1)\left\langle\hat{a}_k^\dagger(t_1)\hat{a}_k(t_2)\right\rangle\right] \\ &+ \left|\left\langle\hat{a}_k^\dagger(t_1)\hat{a}_k^\dagger(t_2)\right\rangle\right|^2 + \left|\left\langle\hat{a}_k^\dagger(t_1)\hat{a}_k(t_2)\right\rangle\right|^2 \\ &+ N_k(t_2)n_k(t_1) + N_k(t_1)n_k(t_2) \\ &+ N_k(t_1)N_k(t_2) + n_k(t_1)n_k(t_2)\end{aligned} \qquad (10)$$

where $n_k(t)$ and $N_k(t)$ are the fluctuations and mean field occupations respectively. The second-order two-times correlations are computed by means of the quantum regression theorem $\langle\hat{o}_1(t_1)\hat{o}_2(t_2)\rangle = Tr\left[\hat{o}_1\hat{U}(t_1, t_2)\{\hat{o}_2\hat{\rho}(t_1)\}\right]$, where $\hat{U}(t_1, t_2)$ is the evolution operator from $t_1$ to $t_2$.

**Simulation parameters.** In the calculations, we use the following values: $U = 1.4$ μeV, $J = 0.4$ meV, $\hbar\Delta_{L,R} = -0.6$ meV, $\kappa = \hbar/\tau = 125$ μeV, $p_L = 50\kappa$ resulting in an initial blueshift $\mu = UN_0 \simeq 0.7$ meV, where $N_0$ is the initial population, and $z(0) = (p_L - p_R)/(p_L + p_R) = -0.66$. These parameters allow us to mimic the period of oscillation, initial blueshift, laser detuning and polariton lifetime observed in the experiment.

**Data availability.** The data that support the plots within this paper and other findings of this study are available from the corresponding authors upon reasonable request.

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

## Acknowledgements

We thank Vincenzo Savona and Christophe Galland for fruitful discussions. The present work is supported by the Swiss National Science Foundation under Project No. 153620 and the European Research Council under project Polaritonics Contract No. 291120.

## Author contributions

A.A. and M.A. carried out the experiment and processed the data. H.F. developed the theoretical model and performed the simulations. F.J. fabricated the sample, A.A., C.O.-P., and G.S. contributed to the sample processing. A.A., M.A., and H.F. prepared the manuscript. M.P.-O. and B.D. supervised the project. All authors contributed to discussions and revised the manuscript.

## Additional information

**Competing interests:** The authors declare no competing financial interests.

