## [Peer Review File · Nature Communications]

REVIEWERS' COMMENTS:

Reviewer #1 (Remarks to the Author):

This manuscript reports the observation of oscillating correlations in a polariton Josephson junction. A theoretical model is presented which agrees well with the data, and thereby provides evidence that these correlations arise from the formation of a squeezed state. I think this is interesting work of significant importance both to the field and the wider community, to my knowledge being the first observation of time-dependent quantum fluctuations in a Josephson junction. I have no technical concerns about this work that would preclude publication. I recommend that the editors accept this paper.

I do have one suggestion for the authors which might improve the manuscript for readers: could they give some results for the magnitude of the squeezing, perhaps as a figure showing how it evolves with time? Time-dependent squeezing is the main effect claimed, and it seems odd not to present this quantitatively.

In the referral the editor asked me to comment on previous reports and specifically the suggestion, by reviewer 3, that the authors investigate how the observed effects depend on detuning. I do not think this is necessary: in my view the data given is sufficient to support the claims. The suggestion to investigate the detuning seems to come from the observation that similar effects might occur in other problems, for example purely photonic systems, and that the detuning dependence would provide a link to such systems. That may be so, but I think it goes beyond the scope of this paper.

Reply to the Referee #1

Referee:

This manuscript reports the observation of oscillating correlations in a polariton Josephson junction. A theoretical model is presented which agrees well with the data, and thereby provides evidence that these correlations arise from the formation of a squeezed state. I think this is interesting work of significant importance both to the field and the wider community, to my knowledge being the first observation of time-dependent quantum fluctuations in a Josephson junction. I have no technical concerns about this work that would preclude publication. I recommend that the editors accept this paper.

I do have one suggestion for the authors which might improve the manuscript for readers: could they give some results for the magnitude of the squeezing, perhaps as a figure showing how it evolves with time? Time-dependent squeezing is the main effect claimed, and it seems odd not to present this quantitatively.

In the referral the editor asked me to comment on previous reports and specifically the suggestion, by reviewer 3, that the authors investigate how the observed effects depend on detuning. I do not think this is necessary: in my view the data given is sufficient to support the claims. The suggestion to investigate the detuning seems to come from the observation that similar effects might occur in other problems, for example purely photonic systems, and that the detuning dependence would provide a link to such systems. That may be so, but I think it goes beyond the scope of this paper.

Authors:

We would like to thank the referee for their very positive comments on the paper.

We also agree with the suggestion of the referee and add the graph with the calculated squeezing magnitude, which is given in the subplot (d) of the Figure 2.